# Combined Free Flaps for Optimal Orthoplastic Lower Limb Reconstruction: A Retrospective Series

**DOI:** 10.3390/medicina59050859

**Published:** 2023-04-28

**Authors:** Pietro G. di Summa, Gianluca Sapino, Daniel Wagner, Michele Maruccia, David Guillier, Heinz Burger

**Affiliations:** 1Department of Plastic and Hand Surgery, Centre Hospitalier Universitaire Vaudois (CHUV), 1011 Lausanne, Switzerland; 2Department of Orthopedic Surgery, Centre Hospitalier Universitaire Vaudois (CHUV), 1011 Lausanne, Switzerland; 3Department of Plastic Surgery and Reconstructive Surgery, University Hospital of Bari, 70100 Bari, Italy; 4Department of Plastic Reconstructive and Hand Surgery, Department of Oral and Maxillofacial Surgery-University Hospital, 21231 Dijon, France; 5Privat Clinic Maria Hilf, 9010 Klagenfurt, Austria

**Keywords:** microsurgery, double free flaps, osteocutaneous reconstruction, orthoplasty

## Abstract

*Background and Objectives*: Open fracture of the lower limb can lead to substantial bone and soft tissue damage, resulting in a challenging reconstructive scenarios, especially in presence of bone or periosteal loss, with a relevant risk of non-union. This work analyzes outcomes of using a double approach for orthoplastic reconstruction, adopting the free medial condyle flap to solve the bone defects, associated to a second free flap for specific soft tissue coverage. Indications, outcomes and reconstructive rationales are discussed. *Materials and Methods*: A retrospective investigation was performed on patients who underwent complex two-flap microsurgical reconstruction from January 2018 to January 2022. Inclusion criteria in this study were the use of a free femoral condyle periostal/bone flap together with a second skin-only flap. Only distal third lower limb reconstructions were included in order to help equalize our findings. Out of the total number of patients, only patients with complete pre- and post-operative follow-up (minimum 6 months) data were included in the study. *Results*: Seven patients were included in the study, with a total of 14 free flaps. The average age was 49. Among comorbidities, four patients were smokers and none suffered from diabetes. Etiology of the defect was acute trauma in four cases and septic non-union in three cases. No major complications occurred, and all flaps healed uneventfully with complete bone union. *Conclusions*: Combining a bone periosteal FMC to a second skin free flap for tailored defect coverage allowed achievement of bone union in all patients, despite the lack of initial bone vascularization or chronic infection. FMC is confirmed to be a versatile flap for small-to-medium bone defects, especially considering its use as a periosteal-only flap, with minimal donor site morbidity. Choosing a second flap for coverage allows for a higher inset freedom and tailored reconstruction, finally enhancing orthoplastic success.

## 1. Introduction

Open fracture of the lower limb can lead to substantial bone and soft tissue damage, resulting in a challenging reconstructive scenario for the plastic surgeon. Extensive trauma and compound fractures may cause bone and periosteal loss, together with soft tissue contamination, finally impacting bone consolidation.

The role of the periosteum in fracture healing is well documented: the absence of periosteum decreases the osteogenetic potential and the ability of callus formation, while increasing infection and non-union rate [1]. Indeed, the lack of bone and periosteal vascular supply (gustillo IIIb-c) are major risk factors for chronic bone infections and non-union [2,3].

The rate of successful non-union management after traditional means of internal fixation and bone grafting ranges from 70 to 92% in the absence of major skeletal loss [4,5]. In cases of poorly vascularized bone beds and open comminuted fractures, achieving bone union may be compromised and further surgery necessary in up to 30% of cases [3].

Moreover, in the context of open fracture or recalcitrant infections with multiple debridements, a cutaneous defect is generally associated with the bone and needs to be addressed. The lack of skin redundancy or vascular jeopardization of local tissue due to trauma, especially on the leg, may remove the possibility of performing a local reconstruction [6,7].

In such rare and challenging cases, we encountered a two-fold problem: on one hand, the non-healing fracture needs to be addressed properly to allow the patient to regain function; on the other hand, the soft tissue reconstruction is mandatory to prevent deep structure exposure and further infections [8].

The vascularized transfer of bone or periosteal segment was identified as a potential solution to bring vascularized structural support to native bony segment. In this context, the medial femoral condyle had a well-defined and easily accessible anatomy: its use is gaining in also popularity in long bone non-union in cases of small-to-medium bone gaps, as described in the previous literature [2].

Despite few options for flaps including (in a composite or chimeric way) bone and skin exist (e.g., osteocutaneous fibula flap, chimeric SCIP flap, chimeric FMC), their use may be limited due to anatomical features and reconstructive inset needs [9].

Hence, the aim of this study is to investigate the outcomes of reconstruction of complex orthoplastic scenarios using a double free flap approach: a free periosteal/bone tissue transfer using a free medial condyle flap, associated with a second skin-only free flap for optimal soft tissue coverage.

## 2. Materials and Methods

All patients who underwent complex two-flap microsurgical reconstruction from January 2018 to January 2022 were included in a retrospective investigation performed on a prospectively maintained database. Inclusion criteria in this study were the use of a free femoral condyle periostal/bone flap together with a second skin-only flap. Only lower limb reconstructions were included in order to help equalize our findings. Out of the total number of patients, only patients with complete pre- and post-operative follow-up (minimum 6 months) data were included in the study. 

Patients’ ages, body mass indexes (BMI) and comorbidities were collected from medical and anesthesiologic charts. Moreover, ethiology of the defect, time between surgery and trauma and orthopedic fixation methods were listed. Operative notes were screened for technique and microsurgical details. Hospital letters and outpatient reports were used to evaluate the hospital stay, the post-operative mobilization protocol, and immediate and late complications. According to the previous literature, major complications were considered partial or total flap loss requiring coverage via a supplementary reconstructive procedure. Minor complications included superficial flap necrosis that did not compromise reconstruction and could be treated using a split thickness skin graft (STSG) or direct closure/flap local advancement. 

X-ray follow-ups were performed at 3 and 6 months to assess periosteal integration and fracture healing, eventually combined with CT scan images.

The study was conducted according to the guiding principles of the Declaration of Helsinki, which was created 1975. Informed consent was obtained from all patients, including approval for photographic/video documentation and reuse of data for scientific publications.

## 3. Results

Seven patients were included in the study: six males and one female. The average age was 49. Among comorbidities, four patients were smokers and none suffered from diabetes. Etiology of the defect was acute trauma in four cases and chronic osteomyelitis in three cases.

Patients’ data and characteristics are outlined in Table 1. One-flap arterial thrombosis occurred in patient 3, requiring take back and exploration, with re-do of proximal anastomosis, leading to a favorable outcome.

No other major complications occurred, and all flaps healed uneventfully. Patients 2 and 6 required redraping at 12 months post-operation due to bulkiness at the ankle/foot level.

Post-operative X-rays confirmed the good ossification of the fracture in all cases. However, in one patient, a secondary arthrodesis at the ankle was required to due to post-traumatic arthrosis.

### 3.1. Surgical Technique

The MFC flap was harvested through an incision on the distal third of the medial thigh. Generally, the MFC flap was harvested under tourniquet homolateral to the defect, allowing a second team to prepare receiving vessels in the leg. After incision at the level of the distal medial thigh, the vastus medialis fascia was incised posteriorly, allowing exposition of the medial femoral condyle and its periosteal blood supply [10]. The dominant vessel—the descending genicular artery—was identified and dissected up to its origin. Among flaps, we had three free periosteal medial femoral condyle flaps and four free cortico-periosteal medial femoral condyle flaps, depending on recipient site requirements.

The skin-only flaps were ALT (anterolateral thigh) flaps in five cases, a DIEP (deep inferior epigastric perforator) flap in one case and a SFAP (superficial femoral artery perforator) flap in one case. Harvesting was performed according to the previous literature [11]. When an ALT flap was raised, this task was, generally, performed contralateral to the limb requiring reconstruction, with the two-fold aim of avoiding possible interference with the tourniquet and sparing the same thigh from a double-flap harvest (ELT + FMC).

The non-union/fracture site was prepared after removal of hardware, with bone resection until effective bleeding. The recipient vessels were identified and protected. Osteosynthesis was performed according to orthopedic needs using the cortico-periosteal medial condyle flap. In case of periosteal-only flaps, the periosteum was bridging the monocortical gap, enveloping cancellous or bone graft. Microsurgery between flaps was performed in a sequential manner in five cases (on the prosecution of the ALT pedicle, the descending branch of the lateral circumflex femoral artery), while in two cases the flaps were anastomosed on two different recipient vessels. Patient and flap data are presented in Table 1.

Post-operatively, all patients were kept in bed for 5 days. Flap monitoring with a handheld doppler was performed every hour for the first 48 h. Weight bearing was allowed according to orthopedic indications.

### 3.2. Case Series

#### 3.2.1. Case 1 (Patient No. 2)

A 65-year-old female was referred to our center because of a chronic osteomyelitis lasting more than 5 years, with a skin defect and chronic fistula to the bone following an open fracture of the leg. The patient had already undergone multiple surgeries with bone debridement and autologous graft unsuccessfully. The patient presented a 4 × 3 cm skin defect with a 6 cm mono-cortical defect on the distal third of the tibial bone, with active secretions. We decided to perform a double free flap using a periosteal-only MFC flap and a DIEP flap for skin coverage (the patient asked for an abdominoplasty at the same time). Microsurgical connection of the MFC were performed end-to-side on the anterior tibial artery, while an end-to-side anastomose to the posterior tibial artery was used for the DIEP flap. The post-operative period was uneventful with no recurrence of the infection. A post-operative X-ray showed a good integration of the periosteal graft. A skin flap liposuction was performed 12 months post-operation. Walking activities could be resumed without support at 2 weeks post-operation with full charge (Figure 1).

#### 3.2.2. Case 2 (Patient No. 6)

A 33-year-old smoker male presented a closed fracture of the I and II metatarsus after a motorcycle accident. The patient underwent emergency dorsal and medial fasciotomies, associated with external fixation. Fasciotomies were treated via negative pressure therapies. Due to jeopardized vascularization after trauma, the orthopaedic team subsequently amputated the thumb, while bone fragments incurred in progressive necrosis, with loss of the first metacarpal base (Figure 2). A double-flap treatment was then planned: this treatment was a cortico-periosteal FMC associated to a coverage ALT flap. The ALT flap was anastomosed to the tibialis posterior vessels, while the FMC was anastomosed sequentially to the ALT pedicle (descending branch). The post-operative course was uneventful with no recurrence of the infection. A post-operative X-ray showed a good integration of the bone vascularized graft at 16 weeks. A skin flap liposuction was performed 12 months post-operation. Walking activities could be resumed after 12 weeks, with progressive charge and physiotherapy until complete recovery (Figure 3).

## 4. Discussion

The use of bone/periosteal flap to improve bone healing was previously assessed in the literature. While the medial femoral condyle flap was initially described as a pedicled periosteal or cortico-periosteal flap for bone reconstruction of the lower limb [12], numerous experimental and clinical studies over the past few decades have confirmed the osteogenic properties of periosteum [13,14]. Recently, via a rodents model, authors have made clear how vascularized periosteal flaps can improve and accelerate allograft–host bone union [15]. Moreover, according to the literature, a similar effect on ossification properties between cortico-periosteal and periosteal-only flaps have previously been confirmed [16]. 

Despite such data, the application of bone/periosteal vascularized transfer in small-to-medium bone gaps of the lower limb (directly due to trauma or after resection of the non-union segment) has not been analyzed in depth. 

Bone resection with osteosynthesis and bone grafting are considered the standard treatments, when facing a bone non-union, for bone gaps up to 6 cm [17]. This approach provides a correct theoretical structure, but the key physiological environment is missing. Therefore, it could be insufficient even in small bone gaps, particularly when bone vascularization is compromised and recurring non-union represents a well-known complication. Poorly vascularized bone bed, previous infection, initially open and severely comminuted fracture or internal fixation with extensive iatrogenic periosteum removal are considered risk factors for surgical failure. 

The induced membrane technique, described by Masquelet et al, can be used to improve local vascularization [18]. However, this technique requires two procedures and does not provide a new tissue specifically intended to allow bone formation.

We, therefore suggest that, even in small bone gaps (mono- or bi-cortical), the use of vascularized bone and/or periosteal transfer: the cortical portion can be tailored according to the gap requiring reconstruction. In the case of free periosteal grafts, this flap can be wrapped around the defect and sutured to itself, providing a proper scaffold with a regeneration chamber that can stimulate physiological pathways and, finally, promote bone union [19]. Indeed, periosteum provides cells for both chondrogenesis and osteogenesis, being involved in all phases of the fracture repair process, and participating in intramembranous and endochondral ossification [20]. This enables both indirect and direct ossification, which we believe is particularly helpful when a bone gap needs to be addressed after high energy trauma and comminute fractures with important periosteal loss, or in chronic osteomyelitis scenarios with the failure of previous of bone healing attempts. Indeed, the use of periosteal flaps for non-union management has been described widely in the literature, mostly in children [21]. Various free and pedicled periosteal flaps without bone graft have been described in skeletally immature patients, showing excellent results [22].

The MFC free flap presents the key advantage as it includes a periosteal layer and cortical bone. It can, therefore, be tailored to different gaps, using its different components to promote bone healing.

In our experience, when raised as periosteal only, this flap can be large enough to easily wrap around bone gap and used as new cortex while the inner bone can be filled with cancellous or synthetic bone. The donor site morbidity is low [23]; the vessel size is rather large and there is no need to sacrifice a major artery.

However, few anatomical characteristics of this flap should be considered as they may harm the reconstruction; these flaws represent the reason for the double-flap approach presented [24]. In fact, although it can be harvested with an osteocutaneous component, the skin paddle is limited in size and has a close association to the bone segment, making difficult for the flap inset and insufficient for coverage of moderate-to-large-sized defects. Moreover, the blood supply of the overlying skin retains some anatomical variations, coming from either the distal cutaneous branch of the DGA or saphenous artery branch (SAB) of the DGA [24]. Notably, in around 20% of cases, it is not possible to harvest both the skin and the periosteal layer on the same pedicle as no cutaneous perforator branches arise from the DGA. Finally, in around 15% of cases, the superior medial genicular artery is the main supply of the medial femoral condyle, with a significantly shorter pedicle length (9.1 vs 4.1 cm) and width (2.1 vs 1.7 mm) compared to the DGA [25]. For all these reasons, we think the free medial condyle could be coupled with a second free flap, which will be designed and chosen specifically to serve the insetting needs. This represents, in our view, a more effective and defensive reconstructive solution.

In a double-flap approach, certain donor site morbidities need to be considered as every harvested flap implicitly brings potential complications at the donor sites. The FMC donor site morbidity was, in our experience, extremely mild. No patients complained of secondary pain at the donor site harvest or experienced secondary fractures when a cortico-periosteal flap was chosen. Concerning the second “coverage” flap, we tried to choose the flap that better fitted anatomical resurfacing needs and matched patient’s body shape. The ALT flap was one of the most common choices for the male population, combining thin tissue, long pedicle to allow ideal inset and, most importantly, a consistent run-off after the perforator, namely the descending branch (DB) of the lateral circumflex femoral artery (LCFA). Notably, the distal caliber of the LCFA resembles the DGA dimensions, making end-to-end anastomosis practical. Nonetheless, in those cases where flaps where not connected sequentially, we could perform end-to-side anastomoses to major axes (both tibialis posterior and anterior) uneventfully. All ALT flaps were purely perforator, lacking significant impact on the quadriceps’ performance.

For the only female patient in the series, a DIEP flap was used. This flap, which is rarely used in our lower limb reconstructive practice, served as the scope for a relevant surface to cover, while at the same time decompressing the anastomotic site with the large skin paddle. Moreover, after CT scan analysis, the resulting DIEP flap was thinner than an ALT flap in the same patient, with an acceptable thickness of 1.5 cm at its edges, which favorably responded to secondary thinning via liposuction, and pleasing aesthetic result. According to the location of the bone defect, the FMC flap can be anastomosed to a run-off of the DIEP flap with a potentially good vessel match. In the described cases, this approach was not possible as the bone defect was too distal to allow for connected flaps; the FMC had to be anastomosed to the tibialis anterior axis.

As it may emerge from our described experience, double-flap reconstructions of the lower limb need precise planning and careful consideration of bone and soft tissue defects prior to reconstruction. Similarly, tight coordination between surgical teams is necessary to avoid majorly impacting the operative time. The “coverage” flap is, generally, harvested while the orthopedics team is involved and performed contralaterally. On the other hand, generally, we harvest the FMC on the thigh homolateral to the defect: this approach allows the placement of the tourniquet and, while a team raises the flap, a second team can prepare the receiving vessels. Such baseline rules can change after AngioCT analysis, if the analysis suggests a significantly more favorable harvesting site (e.g., the absence of a good caliber DGA for the FMC on one side or a favorably placed ALT perforator to optimal flap design). Coordination between teams and meticulous planning were critically to keep operative times in a band ranging from 6 to 8 h.

Moreover, a double-flap reconstruction obviously improves insetting, while lacking the potential limitations due to the connections between the periosteal component and skin branch in the FMC flap. This approach should be particularly indicated for cases of moderate loss of substance of lower limb that require cortico-periosteal bone reinforcement and, at the same time, a solid size skin paddle. When FMC were combined with ALT flaps, they could be anastomosed sequentially to the ALT flaps, using the descending branch of the lateral circumflex femoral artery as a donor vessel. This method can potentially allow us to extend the vascular leash of the bony–periosteal flap and significantly improve insetting freedom to ensure optimal functional outcomes.

## 5. Conclusions

By combining a bone periosteal FMC to a second skin flap for tailored defect coverage, we achieved bone union in all patients despite the lack of initial bone vascularization or chronic infection. In an era where microsurgery is widely available, combination of multiple free flaps can be used to enhance reconstructive efficacy and orthoplastic success.

## Figures and Tables

**Figure 1 medicina-59-00859-f001:**
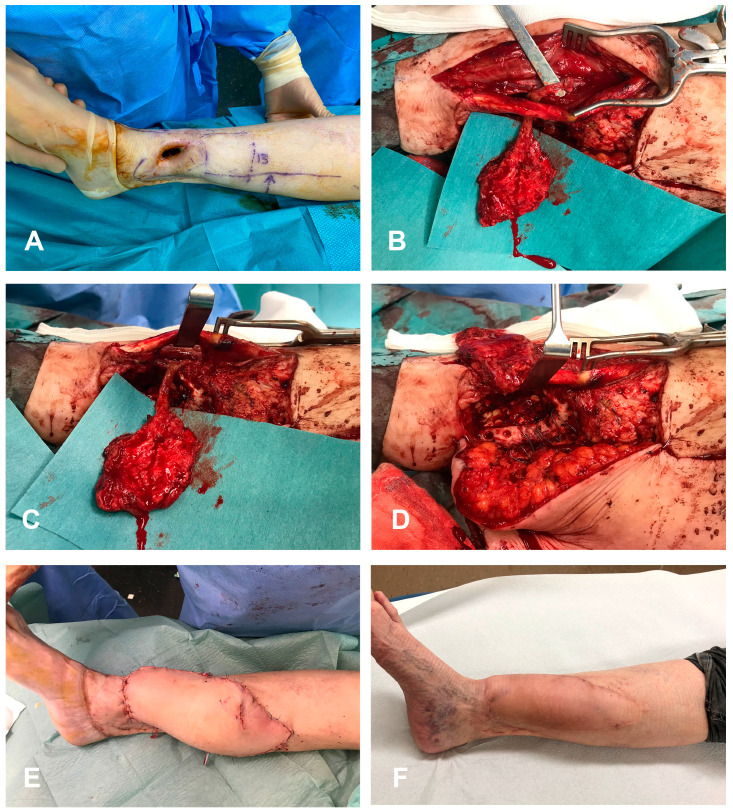
A 65-year-old female was referred to our center because of a chronic osteomyelitis lasting more than 5 years with a 4 × 3 cm skin defect with a 6 cm monocortical defect on distal third of tibial bone, with active secretions (**A**). A double free flap reconstruction was performed, using a periosteal-only MFC flap (anastomosed end-to-side to anterior tibial artery) for bone reconstruction (**B**,**C**). The MFC is adapted to cover the bone defect (**D**). A DIEP flap for skin coverage (anastomosed to the posterior tibial artery in end-to-side manner). Immediate post-operative (**E**) and 3 months follow-up period (**F**) showed an uneventful healing with no recurrence of infection.

**Figure 2 medicina-59-00859-f002:**
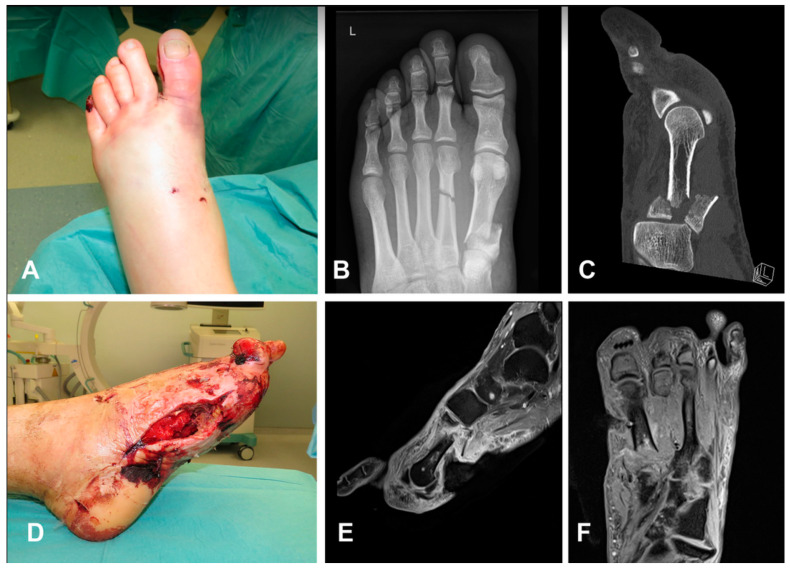
A 33-year-old smoker male presented a closed fracture of I and II metatarsus after a motorcycle accident (**A**–**C**). Patient underwent emergency dorsal and medial fasciotomies, associated with external fixation. Fasciotomies were treated by negative pressure therapies (**D**). Due to jeopardized vascularization after trauma, orthopaedic team subsequently amputated thumb, while bone fragments incurred in progressive necrosis, with loss of first metacarpal base (**E**,**F**).

**Figure 3 medicina-59-00859-f003:**
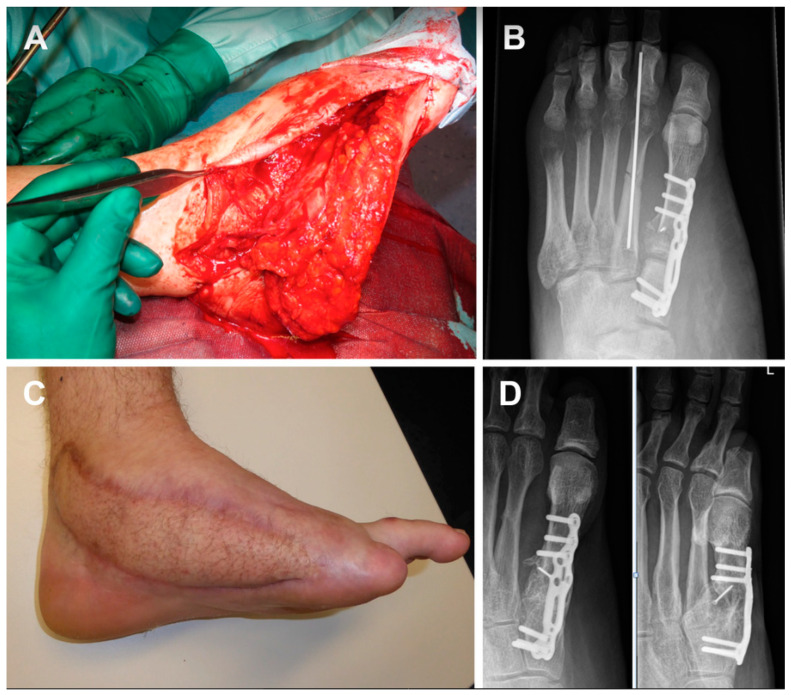
ALT flap was anastomosed to tibialis posterior vessels, while FMC was anastomosed sequentially to ALT pedicle (descending branch) (**A**,**B**). A skin flap liposuction was performed 12 months post-operation (**C**). Post-operative X-ray showed a good integration of bone vascularized graft at 12 months (**D**).

**Table 1 medicina-59-00859-t001:** Patients data and surgical details.

Pt	Anatomic Region	Initial Accident	Active Problem	Bone Defect	Bone Flap	Skin Flap	Skin Flap Size (cm)	Flap Anastomosis	Bone Fixation	Complications	Follow-Up (Months)
1	Ankle	Open fracture	Framework infection and acute OM	Monocortical 4 cm	Periosteal MFC flap	SFAP flap	15 × 6	Single flaps	Periosteal coverage of the bone gap	Secondary arthodesis	18
2	Ankle	Chronic OM	Bone gap following debridement	Monocortical 6 cm	Periosteal MFC flap	DIEP flap	29 × 13	Single flaps	Periosteal coverage the bone gap	None	12
3	Foot	Chronic OM	Bone gap following debridement	Monocortical 3 cm	Periosteal MFC flap	ALT flap	20 × 6	Sequential flaps	Periosteal coverage the bone gap	Arterial Thrombosis and reanastomosis	8
4	Leg	Open fracture	Bone and skin gap	Monocortical 4 cm	Bone MFC flap	ALT flap	21 × 7	Sequential flaps	Plate and screws	None	10
5	Leg	Open fracture	Bone and skin gap	Circulare 1.5 cm	Bone MFC flap	ALT flap	15 × 6	Sequential flaps	Plate and k-wire	None	6
6	Foot	Chronic OM	Bone and skin gap	Circulare 1.5 cm	Bone MFC flap	ALT flap	25 × 8	Sequential flaps	Plate	None	23
7	Foot	Open fracture	Bone and skin gap	Monocortical 4 cm	Bone MFC flap	ALT flap	26 × 8	Sequential flaps	Screws	None	6

ABB: OM, ostheomyelitis; SFAP, superficial femoral artery perforator; DIEP, deep inferior epigastric perforator; ALT, anterolateral thigh.

## Data Availability

The datasets analyzed during the current study are available from the corresponding author on reasonable request.

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
