# Peer review of "Combined Free Flaps for Optimal Orthoplastic Lower Limb Reconstruction: A Retrospective Series"

_medicina, 2023, doi:10.3390/medicina59050859_

Round 1

Reviewer 1 Report

The paper is well written. The study methods are sound.  The findings are well described and the results and interpretations are very useful guides for the microsurgeon who need to find a solution for the osteocutaneous defects that need to be reconstructed
The purpose of the paper was clearly stated by the Authors. Their objective was met through logical processes. The data and statistical analysis were presented well in diagrams and tables. This study will serve as an excellent option in the tool box of the reconstructive surgeons facing complex reconstructions. The limitation of this study is that the study is a retrospective case series and hereby the authors do not compare advantages or disadvantages of alternative flap combinations. I would prefer a more specific discussion regarding donor site morbidity, practical aspects such as two team approach in distant locations, OR time and vascular match in cases of alternative flap choice. Even so, this case series is relevant and will serve as a template for researchers to do extend number of patients and variety of techniques in further studies.

Author Response

Thanks a lot for your comments. Discussion has been enriched treating as requested

  • Donor site morbidity
  • Two team approach
  • OR Time
  • Vascular match in cases of alternative flap choice

A new paragraph now reads

In a double flap approach, certain donor site morbidity needs to be considered as every harvested flap implicitly brings potential complications at the donor sites. FMC donor site morbidity was, in our experience, extremely mild. No patients complained of secondary pain at the donor site harvest, nor experienced secondary fractures when a cortico-periosteal flap was chosen. Concerning the second “coverage” flap we tried to choose the flap that better fitted anatomical resurfacing needs and matched patient’s body shape. ALT flap was one of the most common choices for the male population, combining thin tissue, long pedicle to allow ideal inset and, most importantly, a consistent run-off after the perforator, namely the descending branch (DB) of the lateral circumflex femoral artery (LCFA) . Such run-off has been already described as critical by our team when a flow-through flap is designed1. Noteworthy, the distal caliber of the LCFA resembles the DGA dimensions, making end-to-end anastomosis practical. Still in those cases where flaps where not connected sequentially, we could perform end-to-side anastomoses to major axes (both tibialis posterior and anterior) uneventfully. All ALT flap were purely perforator, without significant impact on the quadriceps performance.

In the only female of the series a DIEP flap was used. This flap, which rarely occurs in our lower limb reconstructive practice, served the scope of a relevant surface to cover, while at the same time decompressing the anastomotic site, with the large skin paddle. Moreover, after CT scan analysis, the DIEP flap resulted thinner then an ALT flap in the same patient, with an acceptable thickness of 1.5 cm at its edges, which favorably responded to secondary thinning by liposuction, with a finally pleasing aesthetic result. According to the location of the bone defect, the FMC flap can be anastomosed to a run-off of the DIEP flap with potentially a good vessel match. In the described cases this was not possible as the bone defect was too distal to allow for connected flaps, and the FMC had to be anastomosed to the tibialis anterior axis.

As it may emerge from our described experience, double flap reconstructions of the lower limb need a precise planning and careful consideration of bone and soft tissue defects to reconstruct. Similarly, tight coordination between surgical teams is necessary not to impact majorly the operative time. The “coverage” flap is generally harvested while the orthopedics team is involved, and is generally performed contralateral. On the other hand we generally harvest the FMC on the thigh homolateral to the defect: this allows the placement of the tourniquet and, while a team raises the flap, a second team can prepare the receiving vessels. Such baseline rules can change after AngioCT analysis, if suggesting a significantly more favorable harvesting site (e.g. the absence of a good caliber DGA for the FMC on one side or a favorably placed ALT perforator to optimal flap design). Coordination between teams and meticulous planning were critically to keep operative time ranging from 6 to 8hrs.

Reviewer 2 Report

Interesting cases, some modifications are mandatory:

-Results: “6 males and 1 females “ —> 1 female

-Table 2 and material and methods: it is unclear which are the 2 compared group in the table! The sentence “Microsurgery details are presented in table 2” is incorrect.

-Discussion: “We therefore suggest, even in small bone gaps (mono or bicortical) the use of ; A vascularized bone and/or periosteal transfer: the periosteum is involved in all phases of the fracture repair process, providing cells for both osteogenesis and..” Problem of paragraph distribution

-Discussion and conclusion: the results present in table 2 are not discussed and therefore conclusion of this manuscript are not based on results from this study

This Manuscript is mainly a case report / case series, this should appear in the title.

Author Response

Interesting cases, some modifications are mandatory:

-Results: “6 males and 1 females “ —> 1 female

Text has been corrected

-Table 2 and material and methods: it is unclear which are the 2 compared group in the table! The sentence “Microsurgery details are presented in table 2” is incorrect.

We apologize for a supplementary table, which was not supposed to be added.

Only table in the paper is table 1 and we apologize for the misunderstanding: this series does not compare two distinct groups. It presents a double flap orthoplastic approach where one of the free flaps is always a FMC flap (in its cortico-periosteal or periosteal only version). All patients followed such reconstructive pattern.

“Microsurgery details are presented in table 2” has been removed. It now reads “Patient and flap data are presented in table 2”

-Discussion: “We therefore suggest, even in small bone gaps (mono or bicortical) the use of ; A vascularized bone and/or periosteal transfer: the periosteum is involved in all phases of the fracture repair process, providing cells for both osteogenesis and..” Problem of paragraph distribution

Thank you for your comments, the paragraph has been reworded. It now reads

We therefore suggest, even in small bone gaps (mono or bicortical) the use of vascularized bone and/or periosteal transfer: the cortical portion can be tailored according to the gap to reconstruct. In case of free periosteal graft only, this can be wrapped around the defect and sutured to itself, providing a regeneration chamber with the proper environment to stimulate physiological pathways to promote bone union 2. Indeed periosteum is involved in all phases of the fracture repair process, providing cells for both osteogenesis and chondrogenesis, participating in intramembranous and endochondral ossification 3.

1

-Discussion and conclusion: the results present in table 2 are not discussed and therefore conclusions of this manuscript are not based on results from this study

Thank you for your comments. Table 2 has now been discussed in the text (both results and a complete new paragraph in the discussion).

Complications have been commented in the results, a new paragraph now reads

“Patients’ data and characteristic are resumed in Table 1. One flap arterial thrombosis occurred in patient 3, requiring take back and exploration, with re-do of proximal anastomosis, with favorable outcome.

No other major complications occurred, and all flaps healed uneventfully. Patients 2 and 6 required redraping at 12 months post-op due to bulkiness at the ankle/foot level. “

The discussion has been expanded to detail and comment outcomes of the series

A new paragraph now reads

“In a double flap approach, certain donor site morbidity needs to be considered as every harvested flap implicitly brings potential complications at the donor sites. FMC donor site morbidity was, in our experience, extremely mild. No patients complained of secondary pain at the donor site harvest, nor experienced secondary fractures when a cortico-periosteal flap was chosen. Concerning the second “coverage” flap we tried to choose the flap that better fitted anatomical resurfacing needs and matched patient’s body shape. ALT flap was one of the most common choices for the male population, combining thin tissue, long pedicle to allow ideal inset and, most importantly, a consistent run-off after the perforator, namely the descending branch (DB) of the lateral circumflex femoral artery (LCFA) . Such run-off has been already described as critical by our team when a flow-through flap is designed1. Noteworthy, the distal caliber of the LCFA resembles the DGA dimensions, making end-to-end anastomosis practical. Still in those cases where flaps where not connected sequentially, we could perform end-to-side anastomoses to major axes (both tibialis posterior and anterior) uneventfully. All ALT flap were purely perforator, without significant impact on the quadriceps performance.

In the only female of the series a DIEP flap was used. This flap, which rarely occurs in our lower limb reconstructive practice, served the scope of a relevant surface to cover, while at the same time decompressing the anastomotic site, with the large skin paddle. Moreover, after CT scan analysis, the DIEP flap resulted thinner then an ALT flap in the same patient, with an acceptable thickness of 1.5 cm at its edges, which favorably responded to secondary thinning by liposuction, with a finally pleasing aesthetic result. According to the location of the bone defect, the FMC flap can be anastomosed to a run-off of the DIEP flap with potentially a good vessel match. In the described cases this was not possible as the bone defect was too distal to allow for connected flaps, and the FMC had to be anastomosed to the tibialis anterior axis.

As it may emerge from our described experience, double flap reconstructions of the lower limb need a precise planning and careful consideration of bone and soft tissue defects to reconstruct. Similarly, tight coordination between surgical teams is necessary not to impact majorly the operative time. The “coverage” flap is generally harvested while the orthopedics team is involved, and is generally performed contralateral. On the other hand we generally harvest the FMC on the thigh homolateral to the defect: this allows the placement of the tourniquet and, while a team raises the flap, a second team can prepare the receiving vessels. Such baseline rules can change after AngioCT analysis, if suggesting a significantly more favorable harvesting site (e.g. the absence of a good caliber DGA for the FMC on one side or a favorably placed ALT perforator to optimal flap design). Coordination between teams and meticulous planning were critically to keep operative time ranging from 6 to 8hrs.”

This Manuscript is mainly a case report / case series, this should appear in the title.

Title has been changed accordingly

“Combined free flaps for optimal orthoplastic lower limb reconstruction: a retrospective series”

Round 2

Reviewer 2 Report

thank you for revision, Article now suitable for publication